Differential responses of hard coral Montipora digitata and soft coral Xenia umbellata to nutrient stoichiometry under heat stress

Mezger Selma D. mezger@uni-bremen.de selma.mezger@web.de
Littke Sophie
Ostendarp Malte
de Breuyn Mareike
Wild Christian
Faculty of Biology and Chemistry, Marine Ecology Group, Universität Bremen , Bremen , Germany
Banaszak Anastazia
Electronic publication date: 2025 Nov 13
Publication date: 2025
Volume: 13
Electronic Location ID: e20273
Received 2025 Feb 12; Accepted 2025 Sep 30
Copyright: ©2025 Mezger et al.
Copyright year: 2025
Copyright holder: Mezger et al.
License: This is an open access article distributed under the terms of the Creative Commons Attribution License, which permits unrestricted use, distribution, reproduction and adaptation in any medium and for any purpose provided that it is properly attributed. For attribution, the original author(s), title, publication source (PeerJ) and either DOI or URL of the article must be cited.
License URL: https://creativecommons.org/licenses/by/4.0/

Keywords: Global stressor, Local stressor, Nutrients, Coral physiology, Climate change, Ocean warming, Phase shift

Funding: The Marine Ecology Department at the University of Bremen, Germany This study was supported by baseline funds of the Marine Ecology Department at the University of Bremen, Germany. The funders had no role in study design, data collection and analysis, decision to publish, or preparation of the manuscript.

==============================
The nitrogen-to-phosphorus (N:P) ratio of dissolved inorganic nutrients may influence coral susceptibility to heat stress, but comparative studies on how different N:P ratios affect soft and hard corals do not exist. This study thus investigated the effects of two different N:P ratios on two common Indo-Pacific reef corals: the pulsating soft coral Xenia umbellata and the hard coral Montipora digitata. Corals were exposed for five weeks to N:P 19:1 (37 µM nitrate, two µM phosphate) and N:P 5:1 (37 µM nitrate, eight µM phosphate), relative to a low nutrient control (N:P 3:1, 0.003 µM nitrate, 0.001 µM phosphate). After two weeks, temperatures were gradually increased from 26 to 32 °C. Nutrient enrichment alone did not affect any response parameter for M. digitata, whereas for X. umbellata, 9% mortality was observed, along with a reduction in pulsation rates (−100% under 5:1 ratio, −58% under 19:1 ratio). Heat stress alone significantly reduced Symbiodiniaceae density and chlorophyll a content of M. digitata, while X. umbellata was unaffected. Combined factors significantly increased mortality for M. digitata (100% and 92%) and X. umbellata (87% and 33%) under the 5:1 and 19:1 ratio, respectively. Symbiodiniaceae density and chlorophyll a content in M. digitata showed the same response as under heat stress alone, while these parameters were unaffected for X. umbellata. Pulsation of X. umbellata was reduced by 100% in both combined treatments. Nutrient enrichment alone likely compromised X. umbellata’s metabolism through the energy-intensive reduction of nitrate to ammonium, halting pulsation. Yet, under combined warming and nutrient addition, X. umbellata showed better fitness than M. digitata, suggesting it may better cope with the predicted environmental changes. Still, both corals were negatively affected, particularly by the 5:1 N:P ratio. The stronger impact of this low ratio likely resulted from exacerbated energy depletion by nitrogen assimilation, while the imbalanced 19:1 ratio may have limited nitrogen assimilation, thereby mitigating stress. These findings suggest that high phosphorus, in combination with high nitrogen, may accelerate coral stress. Hence, tailored nutrient management strategies to aid coral survival in a warming ocean should reduce nutrient concentrations and keep N:P ratios close to the Redfield ratio of 16:1.

Introduction

Coral reefs thrive in nutrient-poor, oligotrophic waters (Odum & Odum, 1955), where their high productivity depends on a mutualistic relationship between the coral host and endosymbiotic dinoflagellates, Symbiodiniaceae (LaJeunesse et al., 2018; Muscatine et al., 1984). This photosymbiosis provides the host with photosynthetically fixed carbon and nitrogen, while the coral offers shelter and nutrients derived from heterotrophic feeding in return (Falkowski et al., 1984; Ferrier-Pagès et al., 2010). The Symbiodiniaceae recycle dissolved inorganic nitrogen (DIN; including ammonium, nitrate, and nitrite) and dissolved inorganic phosphate (DIP), promoting coral growth and survival (Lesser, 2021). The demand of Symbiodiniaceae for nitrogen and phosphorus compounds makes the coral holobiont highly responsive to changes in nutrient availability (Dubinsky & Jokiel, 1994; Fabricius, 2005; Pernice et al., 2012; Stambler et al., 1991). Although corals can restrict the access of nutrients to their Symbiodiniaceae (Jackson et al., 1989; Rands, Loughman & Douglas, 1993), this regulation is limited, as Symbiodiniaceae still respond to nutrient enrichment in the surrounding water (Fabricius, 2005; Pernice et al., 2012; Stambler et al., 1991). Increases in DIN and DIP concentrations can alleviate the nutrient limitations for symbionts, thereby potentially destabilizing the symbiosis, even leading to coral bleaching, and eventually coral mortality (D’Angelo & Wiedenmann, 2014; Shantz, Lemoine & Burkepile, 2016).

In coral reef waters, DIN concentrations typically remain below one µmol/l, peaking at approximately 10 µmol/l, while DIP concentrations range around 0.3 µmol/l, occasionally reaching 0.6 µmol/l (Lesser, 2021). Common DIN:DIP ratios found in coral reef waters, characteristic for pristine reefs, often lie between 4.3:1 and 7.2:1 (Rosset et al., 2017). Significant deviations from these ratios can indicate potential nutrient limitation (Blanckaert et al., 2023). The Redfield ratio of N:P on the other hand, which has been described to be optimal for the growth of phytoplankton, is 16:1 (Redfield, 1958). This indicates a general limitation of N in coral reef ecosystems (Furnas et al., 2005).

Human activities have caused a dramatic increase in nutrient inputs to marine environments, especially since the Industrial Revolution (Krishnamurthy et al., 2010). Anthropogenic contributions now far exceed natural nutrient sources, fundamentally altering marine ecosystems (Bennett, Carpenter & Caraco, 2001; Vitousek et al., 1997). The global input of reactive nitrogen into the biosphere has risen from roughly 15 Tg per year in 1850 to 235 Tg per year by 2020, primarily due to the widespread use of nitrogen fertilizers (Penuelas et al., 2020), along with contributions from municipal wastewater and industrial farm runoff (Barile, 2018; Strokal et al., 2017; Wear & Thurber, 2015), emissions from factories and fossil fuels (Penuelas et al., 2020), and aquaculture operations (Herbeck et al., 2014; Wang et al., 2020).

The additional nitrogen reaches coral reefs through sources such as surface runoff, submarine groundwater discharge, sewage outflows, aquaculture activities, and even natural inputs from fish or seabird colonies (Adam et al., 2021; Graham et al., 2018; Otero et al., 2018; Wear & Thurber, 2015). While natural sources typically supply ammonium, which corals can efficiently use, human-induced enrichment often introduces nitrate (Morris et al., 2019; Shantz & Burkepile, 2014), a less bioavailable form of nitrogen that can intensify stress responses in corals (Burkepile et al., 2020; Fernandes de Barros Marangoni et al., 2020). The overloading of coral reef systems with anthropogenic nutrients has been linked to their degradation, particularly near urbanized or agriculturally developed areas (Brown, 1997; Fabricius, 2005; Wagner, Kramer & Woesik, 2010; Wooldridge, 2009). Thereby, coastal eutrophication decreases water transparency (Cooper et al., 2007), fosters shifts from coral-dominated ecosystems to algal-dominated ones (Adam et al., 2021), reduces habitat complexity (Adam et al., 2021), and disrupts microbial communities (Vega Thurber et al., 2020).

The impacts of nutrient enrichment on coral holobiont physiology were initially debated due to inconsistent results in early experimental studies and as coral reefs span across diverse nutrient environments (Szmant, 2002). However, extensive reviews (D’Angelo & Wiedenmann, 2014; Fabricius, 2005; Morris et al., 2019; Shantz & Burkepile, 2014; Szmant, 2002; Zhao et al., 2021) now provide a clearer mechanistic understanding of how DIN and DIP influence corals, revealing both direct and indirect effects of nutrient enrichment. Nitrogen enrichment has been widely documented to impair coral growth and calcification (Silbiger et al., 2018), reproduction and survival (D’Angelo & Wiedenmann, 2014; Shantz & Burkepile, 2014), reduce thermal tolerance (Wooldridge et al., 2012), lower resilience in specific coral species (Hall et al., 2018), and exacerbate coral disease (Vega Thurber et al., 2014). These adverse effects are often compounded by interactions with additional stressors such as elevated temperatures, ocean acidification, and overfishing, leading to further declines in coral health (Burkepile et al., 2020; DeCarlo et al., 2015; Donovan et al., 2020; Zaneveld et al., 2016). Thermal stress is recognized as one of the primary drivers of coral-Symbiodiniaceae symbiosis breakdown, causing widespread bleaching events (Hoegh-Guldberg, 1999; Hoegh-Guldberg et al., 2007). Elevated nutrient levels, particularly nitrate, can amplify the susceptibility of corals to thermal stress by destabilizing the coral-algae symbiosis (Cunning & Baker, 2012; Shantz & Burkepile, 2014; Wooldridge & Done, 2009).

To add another level of complexity, the stoichiometric ratio of nitrogen to phosphorus (N:P) has emerged as a critical factor in determining coral responses to nutrient enrichment (Morris et al., 2019; Wiedenmann et al., 2013; Zhao et al., 2021). Anthropogenic activities have disproportionately increased nitrogen inputs relative to phosphorus, resulting in unbalanced N:P ratios in marine ecosystems (Mahowald et al., 2008; Penuelas et al., 2020; Seitzinger et al., 2010; Vilmin et al., 2018). Such imbalances can disrupt biological functions in marine organisms, including corals, by destabilizing their photosymbiosis (Ezzat et al., 2015; Morris et al., 2019; Wiedenmann et al., 2013). Phosphate shortages can impair the light-harvesting system (Wiedenmann et al., 2013), restrict cellular growth (Ferrier-Pagès et al., 2016; Wang et al., 2013), inhibit DNA repair during thermal stress (Rodriguez-Casariego et al., 2018), and even render Symbiodiniaceae parasitic, further destabilizing the coral host (Li et al., 2016; Rosset et al., 2017). While balanced nutrient enrichment appears to have minimal impacts on coral thermal tolerance (Courtial et al., 2018; Ezzat et al., 2016b; Hoadley et al., 2016; Tanaka et al., 2014; Wiedenmann et al., 2013), imbalanced N:P ratios may be a key driver of nutrient-enrichment-induced bleaching during thermal stress (Ezzat et al., 2016a; Ezzat et al., 2016b; Rosset et al., 2017; Wiedenmann et al., 2013).

Despite extensive research on nutrient enrichment on coral physiology, most studies have focused on scleractinian corals, with limited attention given to octocorals (see Table S1). A previous study comparing the soft coral Sarcophyton glaucum with the hard coral Turbinaria reniformis, showed that the tested soft coral was more resistant to a change in nutrient concentrations than the scleractinian coral (Blanckaert et al., 2023). These findings emphasize the need for more comparative studies as differential responses to unbalanced DIN:DIP ratios may provide valuable insights into the varying bleaching susceptibilities of corals during thermal stress. Corals with a greater capacity to regulate their internal N:P ratios and maintain balance may demonstrate enhanced resistance to heat stress under such conditions (Blanckaert et al., 2023). Laboratory-based experiments using controlled additions of dissolved inorganic nutrients are a well-established method for investigating these responses under reproducible conditions. Numerous studies have enriched seawater with dissolved inorganic nitrogen and phosphorus compounds to assess coral physiology under controlled conditions (e.g., Blanckaert et al., 2023; Fernandes de Barros Marangoni et al., 2020; Hadjioannou et al., 2019; Mezger et al., 2022; Thobor et al., 2022; Wiedenmann et al., 2013). These approaches enable precise adjustment of nutrient stoichiometry, which is not feasible in field settings, and allow the isolation of individual factors such as temperature and nutrient ratio. Yet, many studies have examined the effects of nutrient enrichment without specifically accounting for the stoichiometric ratio of N:P (see Table S1). However, evidence indicates that N:P ratios play a critical role in shaping coral and algal symbiont physiology, as well as their susceptibility to ocean warming (Wiedenmann et al., 2013). Given the increasing pressures of eutrophication and ocean warming, research that investigates the interactions between these stressors is essential to understanding coral reef resilience under future conditions (Vollstedt et al., 2020; Wiedenmann et al., 2013).

To address these existing knowledge gaps, we conducted a laboratory experiment comparing the responses of a hard coral (Montipora digitata) and a soft coral (Xenia umbellata) to different N:P ratios, both independently and under increased temperatures. Our experimental approach follows established laboratory methodologies for nutrient enrichment studies (e.g., Blanckaert et al., 2023; Fernandes de Barros Marangoni et al., 2020; Hadjioannou et al., 2019; Mezger et al., 2022; Thobor et al., 2022; Wiedenmann et al., 2013), where dissolved nutrient solutions are added to seawater to achieve target stoichiometric ratios. Corals were exposed for five weeks to three different N:P ratios and an incremental temperature increase from 26 °C to 32 °C to assess (1) how replete N:P ratios affect the physiological responses of hard and soft corals, and (2) how these ratios modulate coral responses to thermal stress, aiming to better understand species-specific vulnerabilities in reef ecosystems.

Materials & Methods

Study organisms

The experiment utilized the soft coral Xenia umbellata and the scleractinian coral Montipora digitata. These two taxa were deliberately chosen to represent contrasting functional groups with different characteristics, allowing for a comparative assessment of nutrient stoichiometry effects across coral types. Comparative studies of soft versus hard corals remain rare, despite evidence suggesting that soft corals may respond differently to nutrient imbalance and thermal stress than hard corals (Blanckaert et al., 2023). Despite their differences, both coral species are widely distributed in the Indo-Pacific. M. digitata is especially common along the Queensland coast (Stobart, 1994) and South China Sea (see Jia et al., 2024; Luo et al., 2024), while X. umbellata is common throughout the whole Indo-Pacific region from the Red Sea (Benayahu, 1990) to Australia (Fabricius & Klumpp, 1995) and Indonesia (Fox et al., 2003; Janes, 2013). Colonies of X. umbellata were collected from the northern Red Sea in 2017, while M. digitata colonies, originally sourced from the Indo-Pacific, were obtained from a research group in Wageningen, Netherlands. Specimens of both species were subsequently maintained under stable aquarium conditions (25 °C, salinity ∼35‰, light ∼100 µmol photons m−2 s−1 photosynthetically active radiation (PAR)) in the University of Bremen’s Marine Ecology Department for several years before the start of the experiment.

Corals were fragmented (∼200 fragments per species) 33 days before the experiment. For X. umbellata, 2  ×2  × 2 cm fragments were created using the plug mesh method (Kim, Wild & Tilstra, 2022). Fragments were placed on calcium carbonate plugs (AF Plug Rocks, Aquaforest, Poland) and stabilized with egg crates and mesh. For M. digitata, 1–2 cm fragments were separated using pliers and glued onto calcium carbonate plugs using coral glue (EcoTech Coral glue). Both species’ fragments were transferred to a tower system comprising 12 interconnected tanks for acclimatization under maintenance tank conditions (25 °C, salinity ∼35‰, light ∼100 µmol photons m−2 s−1 PAR) before the experiment.

Experimental design

The five-week experiment (Fig. 1) was conducted in a 12-tank tower system at the University of Bremen, Germany. Each tank (60 L, filled with 40 L seawater) consisted of a holding part for the coral grids and a technical part for equipment which was equipped with heaters (3,613 aquarium heater; 25 W 220–240 V; EHEIM GmbH and Co. Frankfurt, Germany) connected to a temperature controller (Schego TRD digital), pumps (EHEIM CompactOn 1,000 pump; EHEIM GmbH and Co. Frankfurt, Germany) for water flow, and HOBO loggers (HOBO pendant temp/light, Onset, Bourne, MA, USA) for temperature and light intensity monitoring. Tanks were covered with acrylic lids to minimize water evaporation. Lighting was provided by LED modules (Royal Blue and Ultra Blue White WALTRON daytime® LED light, Germany, 12:12 h light:dark) set to match the main holding aquaria with 108 ± 14 µmol photons m−2 s−1 PAR measured under the acrylic lids.

Figure 1 Experimental design of the study including the three different nutrient treatments that Xenia umbellata and Montipora digitata were exposed to, along with the temperature design and measured response parameters.

Asterisks indicate sampling timepoints at the beginning and end of each experimental phase. Created in BioRender. Carvalho, S. (2025) https://biorender.com/k56j267.

Water parameter monitoring (Table 1) included daily measurements of temperature, salinity, and oxygen levels (Hach, HQ40D Portables 2-Kanal Multimeter). PAR levels were measured on days 0, 7, 18, 28, and 35 (LI-COR Data Logger). Weekly tests for KH, Mg, Ca, NO2−, and NH4+ were conducted (TESTLAB MARIN; JBL GmbH & Co. Frankfurt, Germany), with daily pH measurements starting from day 9. To maintain water quality, a 10% daily water exchange was performed. Freshly mixed artificial seawater was added to the technical part of the experimental system, where it was heated to the respective treatment temperature of the day before the circulation pumps were reactivated to mix it with the water in the coral holding part of the tanks. By that, it was ensured that no unintended temperature fluctuations affected the coral fragments during the experiment. Additionally, weekly cleaning of tanks and plugs was conducted to prevent algal overgrowth. Cleaning involved minimal air exposure for coral fragments.

Table 1 Mean values (± SD) of the water quality parameters maintained during the experiment in tanks for each treatment.

Parameters exclude temperature, PO43− and NO3− values.

	Control	5:1 ratio	19:1 ratio	
Salinity (‰)	35.06 ± 0.36	35.10 ± 0.24	35.12 ± 0.21	
PAR (µmol m−2 s−1)	113.66 ± 10.52	105.44 ± 15.88	103.81 ± 13.01	
O2 (mg L−1)	6.58 ± 0.25	6.89 ± 0.37	6.85 ± 0.33	
pH 	8.42 ± 0.07	8.63 ± 0.25	8.59 ± 0.11	
KH (°dH)	9.08 ± 1.04	9.92 ± 1.26	9.71 ± 0.84	
Mg+ (mg L−1)	1,446.67 ± 83.60	1,420.83 ± 75.16	1,435.00 ± 100.71	
Ca2+(mg L−1)	423.33 ± 40.28	429.17 ± 32.13	435. 00 ± 27.23	
NO2− (mg L−1)	<0.01	0,02 ± 0.01	0.03 ± 0.02	
NH4+ (mg L−1)	<0.05	<0.05	<0.05	

Temperatures were maintained at 26 °C for two weeks, then incrementally raised by 2 °C per week until reaching 32 °C. This temperature range was chosen to simulate increasing heat stress above the long-term holding temperature of 26 °C, which served as the temperature control. Additionally, the selected range aligns with previous studies on Xenia spp. as well as other coral taxa, which typically apply temperature increases of no more than + 6 °C above control levels, facilitating cross-study comparability (see Table S1). For the experimental nutrient enrichment, we had three different treatments. Four tanks served as a control with no additional nutrient enrichment (NO3−- = 0.003 (± 0.03) µM and PO43− = 0.001 (± 0.02) µM (N:P ratio ∼3:1)). Four tanks had the “Redfield Ratio” (19:1) treatment aimed for NO3− concentration of 37 µM and PO43− concentration of two µM, concluding in a N:P ratio of 18.5:1, being similar to the Redfield Ratio (Redfield, 1958). The last four tank were exposed to the “Reef Ratio” (5:1) treatment aimed at NO3− concentration of 37 µM and PO43− concentration of eight µM, leading to a N:P ratio of about 4.6:1, which has been reported to commonly be found in coral reefs (Blanckaert et al., 2023). Every day PO43− measurements were carried out using a JBL ProAqua Phosphate Test-Kit and NO3− with a NO3− Sensitiv TestKit (Red Sea Nitrate Pro Test for NO3−). Fluorescence of all samples was then measured using a calibrated photometer (Turner Design Trilogy Laboratory Fluorometer). To reach the targeted concentration, missing amounts were added accordingly with a stock solution of sodium nitrate (eight mM NaNO3, purity > 99%) for NO3− and disodium hydrogen phosphate (two mM PO43−, purity ≥ 98%, SigmaAldrich 71645) for PO43− to the respective treatment tanks.

Physiological analysis

Pulsation rate measurements

Pulsation rates of X. umbellata colonies were monitored on days 0, 14, and 35. Each measurement day, we randomly selected three polyps from a marked colony in each experimental tank and measured pulsation according to the method described by Vollstedt et al. (2020). Measurements were always conducted in the morning to minimize the influence of circadian rhythms. Pulsations were counted over 30 s and then converted to a standard unit of beats per minute (beats min−1). Only complete tentacle contractions (open–fully closed–open) were considered. Average pulsation rates were calculated per colony, providing four independent replicates per treatment.

Symbiodiniaceae density and Chlorophyll a concentration measurements

Coral fragments were sampled on days 0, 14, and 35 with one random colony chosen per species and tank, removed from their plug, rinsed with demineralized water, and frozen at −20 °C until further analysis. Surface area (cm2) was calculated for the normalization of both Symbiodiniaceae density and chlorophyll a concentration measurements. For X. umbellata, the number of polyps in a colony was counted 3 times using tweezers, and the mean number of polyps was then multiplied by an internal surface area lab standard (0.4988 cm2 per polyp), following the method developed by Bednarz et al. (2012). For M. digitata the surface area was determined using the advanced geometry method by Naumann et al. (2009), measuring the length, as well as the minimum and maximum diameter in the center of the respective fragment. If there were several branches in a sample, all were measured individually, and the surface areas were added together. Surface area was then calculated using the following formulas:

(1) r=12∗d

(2) A=2∗π∗h∗r

with d = diameter, A = surface area, h = height and r = radius.

Coral tissue for further analysis was obtained by homogenization in demineralized water using a high-speed homogenizer (VEVOR, FSH-2A) for X. umbellata, and airbrushing of coral fragments with demineralized water (DM) and subsequent homogenization for M. digitata fragments. After homogenisation, two subsamples of two mL each were transferred into Eppendorf cups and centrifuged for 10 min to separate coral tissue from algal cells. Supernatants were discarded and the pellets resuspended in two mL DM and centrifuged for another 10 min. Supernatants were discarded and the pellets frozen until further analysis.

Symbiodiniaceae density was analysed using the haemocytometer counting method (LeGresley & McDermott, 2010). We resuspended the pellets of the subsamples in two mL DM and mixed them thoroughly using a pipette and a vortex. Afterwards, 10 µL were transferred into both chambers of a glass haemocytometer (Improved Neubauer counting chamber, depth 0.1 mm) for two replicate counts of each sample. Counts were then adjusted to the initial sample volume and surface area to determine Symbiodiniaceae density of the colony.

All extraction steps for chlorophyll a (Chl a) analysis were conducted under dark conditions. We resuspended the pellets of the samples in 90% acetone by using a pipette and a vortex to extract the chlorophyll from the algal cells. Samples were then incubated at 4 °C under dark conditions for 24 h. Subsequently we centrifuged samples for 5 minutes and then transferred one mL each into glass cuvettes, allowing for two replicate readings of each sample. Afterwards, we measured Chl a concentrations using a photometer (Turner Design Trilogy Laboratory Fluorometer; excitation: 436 nm; emission: 685 nm) with a respective module for extracted Chl a (non-acidic). Concentrations were standardised to the surface area of the colony and subsequently to the Symbiodiniaceae density to calculate the concentration per algal cell.

Survival analysis

Survival rates were determined daily by counting the number of dead and alive fragments of each species in each tank. Fragments of M. digitata were considered dead when no coral tissue remained on the skeleton. Colonies of X. umbellata were considered dead if no pulsation and tissue necrosis was observed. Dead fragments were subsequently removed from the experiment.

Statistical analysis

The statistical analysis for pulsation rates, Symbiodiniaceae density, and cellular Chl a concentration was carried out using RStudio (Version 2024.09.1) with the packages ggpubr (Kassambara, 2020), rstatix (Kassambara, 2021), tidyverse (Wickham et al., 2019) and emmeans (Lenth et al., 2024). Data normality (Shapiro–Wilk test) and variance homogeneity (Levene’s test) were assessed. As the data did not pass these tests, we analyzed them using the non-parametric Kruskal–Wallis test, followed by post-hoc analysis using a Dunn’s test with Bonferroni adjustment. Survival probabilities were calculated using Kaplan–Meier (KM) curves with the survival package in R (Therneau, 2024). Survival times were compared among treatments using the survdiff function, which performs log-rank tests to assess differences in survival distributions. Pairwise comparisons between treatments were conducted using the pairwise_survdiff function from the survminer package (Kassambara, Kosinski & Biecek, 2024), with p-values adjusted for multiple testing using the Holm–Bonferroni correction method. Results were considered significant with a p-value below 0.05 (p < 0.05).

Results

Effect of nutrient enrichment

In the 1st phase of the experiment (temperature = 26 °C) the effects of nutrient enrichment were investigated independently of additional temperature stress. Under the 5:1 ratio, M. digitata showed no signs of mortality (Fig. 2A), yet Symbiodiniaceae density was significantly increased by 113% (p = 0.021, Fig. 2B, Table S2), while Chl a concentration remained unaffected by nutrient enrichment alone (Fig. 2C, Table S7). X. umbellata colonies showed significantly reduced pulsation (Fig. 3), reaching zero after two weeks of enrichment, compared to the control group (p = 0.01, Tables S11 and S12). Besides this, Symbiodiniaceae density and Chl a concentration in X. umbellata remained unaffected by the 5:1 ratio (Figs. 2B and 2C, Tables S2 and S7).

Figure 2 Effects of nutrient enrichment, heat stress, and combined factors on (A) coral survival probability, (B) Symbiodiniaceae density, and C) Chlorophyll a concentrations.

(A) Survival probability is shown as solid colored lines for each treatment. Different lowercase letters behind each line represent significant differences between treatments. Temperature changes during the experiment are indicated by colored backgrounds and the specific temperature stated at the bottom. (B) Symbiodiniaceae density and (C) Chlorophyll a concentration are shown as mean ± SE (n = 4, except X. umbellata 5:1 on day 35 with n = 2). Uppercase letters above bars indicate significant differences (p < 0.05) between days for the grouped treatments within a species, while lowercase letters in bars indicate significant differences within the respective treatment over time. Coral icons created in BioRender. Carvalho, S. (2025) https://www.biorender.com/wc3hyy4.

Figure 3 Pulsation of X. umbellata shown as mean ± SE.

Biological replicates per sampling timepoint and treatment is n = 4 (exception: 5:1 ratio on day 35 with only two remaining fragments, making statistical testing impossible). Asterisks indicate significant differences (p < 0.05) of the underlying nutrient replete treatment compared to control conditions. Color coded letters indicate significant differences (p < 0.05) within the respective treatment over time.

The 19:1 ratio in the 1st phase of the experiment (temperature = 26 °C) had no significant effect on any parameter of M. digitata. For X. umbellata, the 19:1 ratio reduced pulsation by 58% within two weeks, but this was not statistically significant (p = 0.109, Fig. 3, Tables S11 and S12). X. umbellata also showed some initial mortality but no significant reduction in survival probability (Fig. 2A), while Symbiodiniaceae density and Chl a concentration remained unaffected (Figs. 2B and 2C, Tables S2 and S7).

Effect of temperature increase

The effect of temperature stress alone was analysed by observing the control treatment without nutrient addition during the 2nd phase of the experiment (temperature = 28–32 °C). Survival probability was unaffected by temperature stress for M. digitata (Fig. 2A). However, Symbiodiniaceae density in M. digitata significantly decreased under heat stress (Fig. 2B, Tables S3 and S4), showing a reduction of 71% compared to day 0 (p = 0.019) and day 14 (p = 0.003). Chl a content also significantly decreased with temperature increase, showing an 83% reduction compared to both day 0 and day 14 (p = 0.002, Fig. 2C, Tables S8 and S9). X. umbellata survival probability was unaffected by temperature stress (Fig. 2A), as were pulsation rates (Fig. 3, Table S13). Also, neither Symbiodiniaceae density nor Chl a concentration changed significantly due to heat stress (Figs. 2B and 2C). The comparisons reported here for Symbiodiniaceae density and Chl a content reflect the main effect of temperature across all nutrient treatments, as there were no significant differences detected between different nutrient treatments at each respective experimental day (Tables S1 and S6).

Combined effect of nutrient enrichment and temperature increase

Combined effects of different N:P ratios under heat stress were observed during the 2nd phase of the experiment (temperature = 28–32 °C). The combination of 5:1 N:P ratio and heat stress significantly reduced the survival probability of M. digitata compared to the control (p < 0.001, Table S15), with all corals dying in this treatment before the end of the experiment (Fig. 2A). Regarding Symbiodiniaceae density and Chl a concentration, no significant differences were observed between treatments on day 35, indicating that eutrophication did not exacerbate the temperature-induced effects on M. digitata for these parameters (Tables S1 and S6). Both Symbiodiniaceae density and Chl a concentration were as significantly reduced under heat stress alone as they were under combined stress (Figs. 2B and 2C).

The survival probability of X. umbellata also significantly decreased down to 13% under the combined effect of 5:1 ratio and temperature increase compared to the control treatment (p < 0.001, Fig. 2A, Table S15). Pulsation remained at zero for all fragments under the combined factors (Fig. 3, Table S14). As with M. digitata, there were no significant differences between the eutrophication and control treatments on day 35 for Symbiodiniaceae density and Chl a concentration measurements, suggesting that eutrophication did not exacerbate the effect of temperature alone on X. umbellata (Figs. 2B and 2C, Tables S1 and S6).

When exposed to a N:P ratio of 19:1 under heat stress, M. digitata had a significantly decreased survival probability down to 8% compared to the control (p < 0.001, Fig. 2A, Table S15). The other response parameters showed the same response as under heat stress alone and were not exacerbated by the N:P ratio (Figs. 2B and 2C).

For X. umbellata, a combination of the 19:1 ratio and heat stress significantly decreased their survival probability down to 67% compared to control colonies (p = 0.003, Table S15), less though than under the 5:1 ratio (Fig. 2A, Table S15). While pulsation already started to decline under the 19:1 ratio alone, combined with heat stress, pulsation was significantly decreased (100% reduction, p = 0.012, Fig. 3, Table S14). Symbiodiniaceae density and Chl a concentration in X. umbellata colonies did not differ from control colonies, showing no pronounced negative effects of combined factors (Figs. 2B and 2C).

Differences between coral species

Both M. digitata and X. umbellata had similar Symbiodiniaceae densities and Chl a concentration at the beginning of the experiment (Figs. 4A and 4B, Tables S5 and S10). After two weeks, X. umbellata experienced a significant reduction in Symbiodiniaceae density, with a decrease of 78% (p < 0.001, Tables S3 and 4). This resulted in a significant difference in symbiont densities between the two species (p < 0.001, Table S5). The difference in Symbiodiniaceae density remained throughout the heat stress period, with Montipora digitata maintaining significantly higher symbiont counts compared to Xenia umbellata (113%, p = 0.017, Fig. 4A).

Figure 4 Comparison of (A) Symbiodiniaceae density and (B) Chlorophyll a concentration between coral species.

The boxplots show the median (center line) and the first and third quartiles (lower and upper bounds). Biological replicates per sampling timepoint is n = 12 (exception on day 35: M. digitata n = 6 and X. umbellata n = 10). Asterisks indicate significant differences between species, with *: p < 0.05, **: p < 0.01, and ***: p < 0.001.

While both corals exhibited stable Chl a concentrations during the first two weeks, small differences still resulted in a significant difference in Chl a concentration between the two species on day 14, with M. digitata having a significantly higher concentration (53%, p = 0.043, Table S10). This trend was reversed on day 35, when X. umbellata had significantly higher Chl a concentrations (218%, p = 0.001, Fig. 4B, Table S10).

Survival probability curves for both species revealed a stark contrast (Fig. 2A). M. digitata maintained full survival until reaching 32 °C, at which point it experienced a full collapse: total mortality in the 5:1 ratio and almost total mortality in the 19:1 ratio treatment (8% survival probability). X. umbellata showed some initial mortality within the first two weeks of the experiment, shortly following eutrophication exposure. As temperatures increased, colonies in the 5:1 ratio treatment continued to decline in a dose-dependent manner. Colonies under 19:1 ratio remained a constant survival probability throughout the stepwise temperature increase, until experiencing a significant reduction in survival probability down to 67% near the end of the experiment under 32 °C.

Discussion

This study tested the effects of different N:P ratios on the hard coral M. digitata and the soft coral X. umbellata, both individually and under heat stress. We found that for nutrient enrichment alone only the low N:P ratio increased Symbiodiniaceae density in M. digitata and reduced pulsation rates in X. umbellata, while all other parameters remained unchanged. Combined with heat stress, both N:P ratios significantly decreased the survival probability of both corals, while the high N:P ratio also significantly reduced pulsation rates of X. umbellata.

How do different replete N:P ratios affect the physiological responses of hard and soft corals?

Nutrient enrichment alone did not cause mortality in M. digitata and X. umbellata

Regardless of N:P ratio, none of the hard coral fragments died during the experiment. This aligns with earlier studies that reported no significant mortality under nutrient enrichment alone (Fabricius et al., 2013; Kline et al., 2006; Kuntz et al., 2005). While some studies reported mortality in hard corals exposed to nutrient enrichment, they did not report whether these rates were significantly different from controls (Renegar & Riegl, 2005; Samlansin, Chawakitchareon & Rungsupa, 2020). For X. umbellata, initial mortality occurred after two weeks of nutrient enrichment in both N:P treatments, though it was not significant. Similar results were obtained by Thobor et al. (2022), who reported initial mortality in X. umbellata exposed to 37 µM nitrate for three weeks, without significant differences from controls.

Low N:P ratio significantly increased Symbiodiniaceae density in M. digitata, while high N:P ratio did not affect algal symbionts

While no mortality was expected, we expected nutrient enrichment to increase Symbiodiniaceae densities and Chl a concentrations, as previous studies observed such responses at moderate to high dissolved inorganic nitrogen (DIN) levels (Nalley et al., 2023). However, in our study, only the low N:P ratio (5:1) significantly increased Symbiodiniaceae density in M. digitata, while the high N:P ratio (19:1) showed no notable effect on either Symbiodiniaceae or Chl a levels after two weeks. Studies using similar low N:P ratios (between 3−7.5:1) report mixed outcomes regarding their impact on Symbiodiniaceae and Chl a (Béraud et al., 2013; Blanckaert et al., 2023; Ezzat et al., 2015; Ezzat et al., 2016b; Stambler et al., 1991), indicating that a low N:P ratios alone may not be a strong driver of this coral physiological response. Conversely, research using high N:P ratios (16–22:1) often observed increased Symbiodiniaceae density and Chl a concentrations (Hall et al., 2018; Rosset et al., 2017; Tanaka et al., 2014; Wiedenmann et al., 2013), but these studies typically involved longer exposure periods (4–6 months). This discrepancy suggests that short-term experiments, like ours, may be insufficient for detecting such changes.

Besides looking at the N:P ratio, the nitrogen source itself can have significantly different effects. Fernandes de Barros Marangoni et al. (2020) observed no significant effect on Symbiodiniaceae density or Chl a at a 17:1 N:P ratio using nitrate for 5 weeks, though ammonium enrichment in the same ratio caused substantial increases. Generally, studies testing ammonium consistently reported a stronger response than nitrate, increasing the amount of Symbiodiniaceae and overall Chl a concentrations (Béraud et al., 2013; Ezzat et al., 2015; Fernandes de Barros Marangoni et al., 2020; Muller-Parker et al., 1994; Muscatine et al., 1989; Stambler et al., 1991). This highlights the importance of the nitrogen source, as ammonium is more bioavailable and directly assimilated by both corals and Symbiodiniaceae, enhancing photosynthesis and growth (Nalley et al., 2023; Shantz & Burkepile, 2014). Nitrate, in contrast, must first be reduced to ammonium, an energy-intensive process that may limit its physiological impact (Dagenais-Bellefeuille & Morse, 2013; Patterson et al., 2010).

Another important factor to look at, in addition to N:P ratio and nitrogen source, is the actual nutrient concentration in the water. Regarding nitrate concentrations, studies with low nitrate concentrations (3–7 µM) typically report no significant changes in Symbiodiniaceae density or Chl a (Blanckaert et al., 2023; Dobson et al., 2021; Fernandes de Barros Marangoni et al., 2020; Rosset et al., 2017). Higher concentrations (>20 µM) can induce increases, though this is not consistent across studies (Marubini & Davies, 1996; Schlöder & D’Croz, 2004). Importantly, nutrient uptake rates appear highest when DIN is paired with dissolved inorganic phosphorus (DIP) (Blanckaert et al., 2023). This could explain the increased Symbiodiniaceae density observed under the low N:P ratio in our study, where phosphate was less limiting compared to the high N:P ratio, allowing for higher nutrient uptake rates of the corals in the 5:1 ratio treatment.

The duration of exposure may also play a role. Longer experiments (e.g., 30–40 days) with comparably high nitrate concentrations have shown stronger responses (Marubini & Davies, 1996; Schlöder & D’Croz, 2004). Nevertheless, a meta-analysis by Nalley et al. (2023) found that exposure duration was not a significant predictor for most coral response parameters, except for growth and survival, suggesting our two-week period should have been sufficient.

Overall, the limited response of M. digitata to nutrient enrichment in our study likely reflects a combination of N:P ratio, nitrogen source, concentration, and exposure duration. These findings underscore the need for further research to clarify the interplay of these factors in coral nutrient dynamics on hard corals.

Nutrient enrichment alone did not affect algal symbionts of X. umbellata

Compared to studies on hard corals, few experimental studies have explored the effects of nutrient enrichment on soft corals, and even fewer report specific N:P ratios (Table S1). Blanckaert et al. (2023) investigated a low N:P ratio (3:1) using nitrate in Sarcophyton glaucum and found no significant effects on Symbiodiniaceae density or Chl a concentration after 5 weeks, aligning with our findings for X. umbellata at a 5:1 ratio. Similarly, experiments on X. umbellata with isolated nitrate (Thobor et al., 2022) or phosphate enrichment (Klinke et al., 2022) also reported no significant changes in algal symbionts, despite vastly different N:P ratios (200–1200:1 and 0.06–0.5:1, respectively). These results suggest that the N:P ratio alone may not strongly influence algal symbionts in X. umbellata under nutrient enrichment.

Soft corals exhibit lower nitrogen and phosphorus uptake rates compared to scleractinian corals, even under high nutrient concentrations (Blanckaert et al., 2023). This limited uptake likely minimizes nutrient accumulation in soft coral tissue, preventing significant shifts in host-symbiont interactions or elemental stoichiometry (Blanckaert et al., 2023). Consequently, soft corals may be less sensitive to reef nutrification than hard corals (Baum et al., 2016; Schleyer & Celliers, 2003). However, studies using ammonium as the nitrogen source (Bednarz et al., 2012; McCauley & Goulet, 2019) reported increases in Symbiodiniaceae density and Chl a concentration in soft corals after 1–4 weeks at 10–20 µM NH4. This suggests a nitrogen source-dependent effect, with the higher bioavailability of ammonium facilitating more efficient uptake by algal symbionts and stimulating growth (Nalley et al., 2023; Shantz & Burkepile, 2014). In contrast, the lack of response in nitrate-based studies may reflect slower assimilation or energy constraints associated with nitrate reduction (Dagenais-Bellefeuille & Morse, 2013; Patterson et al., 2010).

Pulsation of X. umbellata was significantly decreased under low N:P ratios

In X. umbellata, the 19:1 N:P ratio reduced pulsation by 58% after two weeks, though not significantly. This aligns with Thobor et al. (2022), who observed a 34% reduction under 37 µM nitrate enrichment over two weeks, also without statistical significance. However, in our 5:1 treatment, pulsation rates dropped to zero, showing a significantly stronger response than Thobor et al. (2022), which focused on high N:P ratios (200–1200:1) from nitrate enrichment alone.

A low N:P ratio, such as 5:1, may induce relative phosphorus sufficiency, promoting Symbiodiniaceae proliferation more than higher, imbalanced ratios (Ezzat et al., 2015). Although this initial proliferation is not directly reflected in our measured symbiont densities, this is likely due to the first sampling point occurring at day 14. It is possible that symbiont density initially increased early in the exposure period but was subsequently reduced through partial symbiont expulsion as a host response. The possible proliferation of Symbiodiniaceae in this treatment might have decreased carbon translocation to the coral host, exacerbating energy depletion compared to nitrate enrichment alone (Ezzat et al., 2015). Furthermore, nitrate reduction to ammonium is energy-intensive, consuming electrons and resources (Dagenais-Bellefeuille & Morse, 2013), potentially contributing to the observed cessation of pulsation, an energetically costly process (Kremien et al., 2013). This underscores functional trade-offs in soft corals, where energy allocation shifts away from non-essential processes like pulsation to vital survival mechanisms, also emphasizing the need for more comparative research on hard and soft coral responses to nutrient enrichment.

In summary for nutrient enrichment alone: While significant changes were observed primarily under the low N:P ratio, the overall responses were subtle, with no clear tendencies across all parameters. Yet, these findings further support that nutrient enrichment has the potential to disrupt the coral-algal symbiosis and reduce energy allocation to the coral host (Blanckaert et al., 2023; Dagenais-Bellefeuille & Morse, 2013; Ezzat et al., 2015). The significant changes under the low N:P ratio may reflect the complete absence of nutrient limitation in this treatment, allowing for greater disruption of the symbiosis and leading to early signs of stress. Further research is needed to better understand the role of N:P ratio deviations in coral stress responses under nutrient-rich conditions.

How do different replete N:P ratios modulate coral responses to thermal stress?

Thermal stress alone caused bleaching in M. digitata but not in X. umbellata

Temperature alone led to a significant reduction in both Symbiodiniaceae density and cellular Chl a concentration in the hard coral M. digitata. This is consistent with the common bleaching response of hard corals to temperature stress (Douglas, 2003; Fitt et al., 2000), characterized by the loss of color due to the partial elimination of Symbiodiniaceae and the degradation of their algal pigments. In contrast, for X. umbellata, the addition of warming did not significantly influence Symbiodiniaceae density or cellular Chl a concentration. This indicates that the photophysiology of X. umbellata was not affected by thermal stress, which may be due to the origin of the colonies from the Red Sea, a region known for corals exhibiting high thermal tolerance (Fine, Gildor & Genin, 2013). Additionally, the ability to pulsate allows soft corals of the family Xeniidae to enhance water mixing across the coral-water boundary layer (Kremien et al., 2013), potentially helping to mitigate increases in water temperature near the coral in a warming scenario.

Combined factors significantly decrease survival in M. digitata regardless of N:P ratio

In our study, both N:P ratios combined with elevated temperature stress significantly reduced survival in M. digitata. In both eutrophication treatments we observed increased algal growth, a potential additional stress factor facilitating coral mortality (Smith et al., 2006). Only two studies have explicitly reported hard coral mortality under combined nutrient enrichment and temperature stress. Most studies without mortality data likely observed no coral death during their experiments. Of the two available studies, Fabricius et al. (2013) reported no effect of single or combined stressors on coral mortality. However, their experiments featured prolonged exposure to lower nitrate levels (four µM), suggesting that nutrient concentration plays a crucial role in survival outcomes. Similarly, most studies reporting no mortality often used nitrogen concentration below six µM. Conversely, Wiedenmann et al. (2013) observed 100% mortality of coral fragments exposed to a 42:1 ratio under combined stress, compared to no mortality at a 21:1 ratio, both using 6.5 µM nitrate. This underscores the additional critical role of N:P ratios in coral survival. Surprisingly, in our study, M. digitata experienced complete mortality under the 5:1 ratio and nearly complete mortality under the 19:1 ratio, indicating no significant differences between these ratios. While oligotrophic reefs typically exhibit natural N:P ratios between 4.3:1 and 7.2:1 (Rosset et al., 2017), ambient nutrient concentrations in such systems are extremely low, often in the sub-micromolar range (Lesser, 2021). The higher nutrient concentrations in our study likely induced nitrogen toxicity, causing coral collapse regardless of the N:P ratio. These findings emphasize that not only the N:P ratio but also absolute nutrient concentrations must be considered when evaluating coral responses to combined stressors.

Symbiodinaceae density and Chl a response of M. digitata is driven by temperature stress

In M. digitata fragments surviving the combined nutrient enrichment and heat stress, Symbiodiniaceae density and Chl a concentrations significantly decreased, yet the nutrient enrichment did not exacerbate the temperature-induced effects. This aligns with several laboratory studies, which also found temperature-driven changes in these parameters without exacerbation or mitigation by nutrient enrichment (Béraud et al., 2013; Higuchi, Yuyama & Nakamura, 2015; Hoadley et al., 2016; Schlöder & D’Croz, 2004; Tanaka et al., 2014). The absence of a synergistic effect with nutrient enrichment suggests that nutrient levels alone do not significantly influence temperature-driven physiological or photosynthetic stress in Symbiodiniaceae and that temperature stress appears to be the primary driver of changes in these parameters. Furthermore, several studies report no differences between nutrient enrichment alone and combined nutrient enrichment with heat stress in response parameters (Ezzat et al., 2016b; Hadjioannou et al., 2019; Hall et al., 2018). This further supports the idea that nutrient enrichment does not directly exacerbate physiological stress in corals under combined stress conditions.

Combined factors significantly decrease survival X. umbellata, but less under high N:P ratios

Compared to hard corals, only three studies have investigated the combined effects of inorganic nutrient enrichment under temperature stress on soft corals (Klinke et al., 2022; Mezger et al., 2022; Thobor et al., 2022). Our study found significantly increased mortality of X. umbellata fragments under both N:P ratios tested. This aligns with previous findings where temperature alone did not cause mortality in X. umbellata (Klinke et al., 2022; Thobor et al., 2022), but combined nitrate enrichment reduced survival to 74% of the fragments (Thobor et al., 2022). In our experiment, survival probability was even lower, at 67% in the 19:1 ratio and only 13% in the 5:1 ratio treatment. These differences in survival may be explained by varying N:P ratios across studies. While nutrient uptake rates increase with higher seawater concentrations (Aubriot & Bonilla, 2018; Lubsch & Timmermans, 2020; Perini & Bracken, 2014), the limitation of one nutrient restricts the assimilation of the other, as seen with combined DIN and DIP enrichment (Blanckaert et al., 2023). Therefore, the corals in Thobor et al. (2022), using the same nitrate concentration but in a highly imbalanced N:P ratio (200-1200:1), experienced the lowest mortality, followed by the 19:1 ratio in our study, and the 5:1 ratio being the most detrimental. This suggests that higher, more imbalanced N:P ratios at similar nitrate concentrations could reduce the stress caused by nutrient enrichment.

Combined stress significantly reduces pulsation rates of X. umbellata under high N:P ratio

Under combined stressors, we did not observe any changes in Symbiodiniaceae density or Chl a concentration for X. umbellata, consistent with the results of Thobor et al. (2022). Pulsation rates were reduced to zero in the combined treatment, similar to the findings of Thobor et al. (2022), regardless of the N:P ratio. However, it is noteworthy that in our study, pulsation under the 5:1 ratio was already reduced by 100% under pure nutrient enrichment, meaning the addition of temperature stress could not exacerbate this effect. This early collapse of pulsation under the 5:1 ratio could serve as an indicator of the higher stress levels in X. umbellata exposed to this treatment compared to the 19:1 ratio, as corals in the 5:1 ratio showed significantly lower survival probability under combined stressors. It has been suggested that pulsation rate can be used as a non-invasive, easily detectable, and inexpensive early warning indicator for changes in water quality and ocean warming (Vollstedt et al., 2020), and our study further supports this parameter as a valuable early bioindicator. In contrast, under the 19:1 ratio, nutrient enrichment alone reduced pulsation, but not significantly; however, under combined stress, pulsation was significantly reduced by 100%. This may be explained by potentially different nitrate uptake rates, resulting in lower stress in the high N:P ratio treatment alone (Blanckaert et al., 2023), which, when combined with temperature stress, led to a collapse of pulsation by the end of our experiment.

Conclusion

In summary, combined nutrient enrichment and temperature stress significantly reduced survival in both M. digitata and X. umbellata, especially under low N:P ratios. For the hard coral M. digitata, our findings suggest that both the N:P ratio and nitrate concentration play critical roles in coral survival, expanding upon the insights provided by Fabricius et al. (2013) and Wiedenmann et al. (2013). For the soft coral X. umbellata, N:P ratio appears to play a more significant role than nutrient concentration alone, as seen in comparisons to Thobor et al. (2022). Overall, X. umbellata was less affected by nutrient enrichment under combined temperature stress, highlighting its greater tolerance to environmental stressors compared to hard corals (Thobor et al., 2022; Vollstedt et al., 2020). This resilience may be attributed to lower nitrogen and phosphorus uptake rates (Blanckaert et al., 2023), which make soft corals less responsive to nutrient enrichment (Baum et al., 2016; Schleyer & Celliers, 2003). Soft corals are also known for their ability to rapidly colonize new areas due to high fecundity and unique dispersal modes (Benayahu & Loya, 1985; Fox et al., 2003). Members of the Xeniidae family, in particular, are recognized as invasive species that dominate alternative reef states around the world (Baum et al., 2016; Fox et al., 2003; Mantelatto et al., 2018; Menezes et al., 2021; Ruiz-Allais, Benayahu & Lasso-Alcalá, 2021). This aligns with observations of phase shifts from hard-coral to soft-coral-dominated reefs (Norström et al., 2009; Reverter et al., 2022), especially under future ocean conditions. Supporting hard coral resilience in a changing climate requires urgent action. While addressing greenhouse gas emissions remains paramount, proactive interventions are also critical (Peixoto, Sweet & Bourne, 2019). Managing local stressors, such as inorganic eutrophication, through targeted nutrient management strategies could play a key role in promoting hard coral survival in warming oceans.

Supplemental Information

Supplemental Information 1 Published studies focusing on nutrient enrichment on coral physiology looking at similar response parameters

Supplemental Information 2 Raw data used for statistical analysis and figures

Supplemental Information 3 Statistical results tables

Supplemental Information 4 Codebook

We would like to thank Arjen Tilstra and Verena Wutz for technical support in the implementation of this experiment. We want to especially thank Svea Vollstedt for her support in experimental design and execution, daily marine aquaria maintenance activities during the experiment, as well as sample processing. We would also like to thank Susana Carvalho for preparing Fig. 1 and the coral icons for Fig. 2 of this manuscript using BioRender.

We acknowledge the use of ChatGPT (OpenAI) for text refinement and clarity improvements. This does not include generative editorial work or autonomous content creation.

Additional Information and Declarations

Competing Interests

Author Contributions

Data Availability

The authors declare there are no competing interests.

Selma D. Mezger conceived and designed the experiments, performed the experiments, analyzed the data, prepared figures and/or tables, authored or reviewed drafts of the article, and approved the final draft.

Sophie Littke performed the experiments, analyzed the data, prepared figures and/or tables, authored or reviewed drafts of the article, and approved the final draft.

Malte Ostendarp performed the experiments, authored or reviewed drafts of the article, and approved the final draft.

Mareike de Breuyn performed the experiments, authored or reviewed drafts of the article, and approved the final draft.

Christian Wild conceived and designed the experiments, authored or reviewed drafts of the article, and approved the final draft.

The following information was supplied regarding data availability:

The raw data is available in the Supplementary Files.

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
