# Peer review of "Differential responses of hard coral Montipora digitata and soft coral Xenia umbellata to nutrient stoichiometry under heat stress"

_PeerJ, doi:10.7717/peerj.20273_

## Round 0.1 · original submission · Major Revisions

· Academic Editor

Major Revisions

Three expert reviewers have evaluated your manuscript and their comments and suggestions can been seen below. The research questions are relevant and fill a knowledge gap. However, the manuscript needs to be reworked to improve clarity in the methods used as well as an improved results section. The reviewers have suggested ways to improve the flow of that section, which is currently a but disorganized. Also please ensure that the discussion is presented in a manner coherent to the new results section. The reviewers have also pointed out how to improve the discussion.

Prior to resubmission, please ensure that you respond to each point made by the reviewer and clearly show where modifications have been made. If you disagree with any of the comments, provide a clear justification.

Reviewer 1 ·

Basic reporting

The manuscript by Mezger et al. investigates the effects of nutrients and temperature on two coral species—one soft coral and one hard coral. The topic is important; however, the manuscript requires major revisions. The results section is too disorganized and does not effectively present the findings. It should be rewritten for clarity. Additionally, the discussion lacks coherence, and each paragraph should have a clear message, which is currently missing.

Experimental design

Line 52-53: The phrase "This capacity ..." is unclear and should be reworded for better comprehension.
Materials and Methods: The selection of coral species needs further explanation. Why was one species chosen from soft corals and the other from hard corals? Please clarify the rationale behind this selection.
Line 166: The term "preparation" is vague. Do you mean fragmentation? Also, what were the maintenance conditions before the experiment started? Please provide more details.
Line 182-183: Specify the light intensity used during the experiment.
Line 189: Did you measure PAR under the acrylic lid? The lid may have affected light intensity and created variations between aquaria. Please clarify.
Line 220-229: The method used to measure surface area is not well-suited for accuracy, particularly since the two species have different morphologies. Please provide a reference for the chosen method and explain why more accurate alternatives were not used.

Validity of the findings

The results section is disorganized, with different parameters mixed together. It should be rewritten to follow the order of the Materials and Methods section and present each parameter separately.
Line 303: The reported 71% decrease should specify which species it refers to.

Additional comments

The sections Surface Area Measurements, Tissue Homogenization, and Symbiodiniaceae Density Measurements should be combined into a single section titled Symbiotic Algal Density Measurement for better readability.
The discussion lacks clear conclusions for each paragraph. Each paragraph should have a distinct message, which is currently unclear. The discussion needs to be restructured for better coherence.

Reviewer 2 ·

Basic reporting

The article identifies nutrient enrichment using scientific solutions for coral at various water temperatures. The study indicates the importance of these types of experiments in the era of global burning. However, the introduction of the nutrient using the scientific solution could increase the ethical issue; please clearly state if there any previous articles has been done the same. Additionally, please explain why you
used this method

Experimental design

The author stated that there is a 10 % daily water exchange, which the reviewer expected when the water temperature changes in the current experimental design. Has the same amount of water been stored as in the water in the experimental water system? Please clearly state the water changes that impacted the temperature

There is a selection of water temperatures that need to be explained, from 26 to 32 degrees Celcius. Is there any reason? How about 34 degrees Celcius? Please explain why you used this method

Validity of the findings

no comment

Additional comments

no comment

Reviewer 3 ·

Basic reporting

The authors did a great job with the introduction. There is a great background on the importance of N:P ratios and nutrient enrichment on coral reefs, and the supplementary table containing a review of articles relevant to this study is especially beneficial. It is very helpful for readers to provide context to the gaps in knowledge, particularly the columns summarizing physiological responses to different N:P ratios.

The structure of the manuscript is consistent throughout and is clearly delineated. The tables are clear, including the raw data with the defined columns in the code book document. The figures are clean and well-made. However, I have suggestions for the order in which the figures are presented that I believe would help to improve clarity when reading the results section.

In the initial reading of the results, I found it hard to look at 3 different figures to find the results for M. digitata over different physiological responses ( mortality, symbiont density, chl a concentration) and while looking within the first phase of the study (26ºC before temperature was increased), and then also to find the significant differences (lowercase letters were “hidden” in the bars for the specific pairwise comparison in figure 4A). For clarity, I recommend making and new Figure 2 as one large figure that combines: Row 1: figure 2 A and B, Row 2: figure 4 A, Row 3: Figure 5A, and making sure that the experiment phases line up along the x-axis. Then it is easier for the reader to compare the changes in physiological responses between the experimental phases. Then, a new figure 3 which would be figure 4B + 5B, and figure 4 would become the pulsation figure (currently figure 3).

My last feedback to improve clarity of the manuscript is regarding the statistical testing and reporting for pairwise significant differences. I would make it clearer in the figure captions what the capital versus lowercase letters are referring to for the statistical comparisons. This may also be improved by providing tables with the statistical results, so that the readers can refer to those for specific pairwise comparisons more clearly.

Overall, the results are relevant to the hypothesis.

Experimental design

The research questions are very relevant and meaningful, highlighted by the supplementary table that the authors generated with a literature review of previous nutrient and thermal stress studies with hard and soft corals. This manuscript clearly addresses the knowledge gaps.

The methodology presented in the manuscript is thorough, well-written, and scientifically rigorous. The authors provided excellent detail for replication by other investigators.

Validity of the findings

The results presented in this manuscript are scientifically supported and in-line with previous findings, which the authors detail in the discussion. The authors also state how their new findings build upon previous hypotheses regarding the importance of nitrogen source and the N:P ratio itself in influencing coral physiological response. The conclusions the authors make in this manuscript are supported by their findings.

Additional comments

My additional comments are minor suggestions for improving clarity, particularly in the results section and figures.

For the figure 3 caption, because the statistical testing for the 5:1 ratio on day 35 did not have enough replication, then I would remove the "(b)" and grey asterisk, which I think are indicating that this test was significant.

For figures, I would suggest that the authors make different asterisks based on different levels of p-value significance: P<0.05 = * , P<0.01 = ** , P<0.001 = ***
When referring to the experiment phases in the text, I would put for 1st phase (temp = 26ºC), and 2nd phase (temp = 28-32ºC).

Line-by-line comments:

Line 223: “Surface area (cm) was calculated for normalization (of?) Symbiodiniaceae density.” Grammar

Line 293: I would state that this was specifically during phase 1 when looking at nutrient enrichment alone.

Lines 302-304: is this significant reduction in symbiont density looking at figure 4A, and comparing day 35 (32ºC) versus days 0 and day 14? And so the significant p-values being referred to in the text correspond to the letters above the bars? If the p-values are referring to specific pairwise comparisons, then there should be the same lowercase letters in the bars as in for the 5:1 ratio in figure 4A. The same goes for lines 304-305 and 307-308. To clarify, I would refer to either Figure 4A/5A or 4B/5B in the text.

Lines 473-474 – I agree with the authors’ suggestion about initial symbiont proliferation due to low N:P, but this is not reflected in their results about symbiont density, likely because the first time point of measurement is day 14. Does this suggest then that the X. umbellata had an initial increase in symbiont density, followed by symbiont expulsion due to host response?

---

## Round 0.2 · accepted · Accept

· Academic Editor

Accept

Thank you for your detailed responses to all of the reviewer comments. I am satisfied with the changes that have been made to the manuscript and am proposing that it be accepted for publication.